# Improving Efficiency: Automatic Intelligent Weighing System as a Replacement for Manual Pig Weighing

**DOI:** 10.3390/ani14111614

**Published:** 2024-05-29

**Authors:** Gaifeng Hou, Rui Li, Mingzhou Tian, Jing Ding, Xingfu Zhang, Bin Yang, Chunyu Chen, Ruilin Huang, Yulong Yin

**Affiliations:** 1CAS Key Laboratory of Agro-Ecological Processes in Subtropical Region, Hunan Provincial Key Laboratory of Animal Nutritional Physiology and Metabolic Process, Hunan Research Center of Livestock and Poultry Sciences, South Central Experimental Station of Animal Nutrition and Feed Science in the Ministry of Agriculture, National Engineering Laboratory for Poultry Breeding Pollution Control and Resource Technology, Institute of Subtropical Agriculture, Chinese Academy of Sciences, Changsha 410125, China; hougaifeng@isa.ac.cn (G.H.); lirui@isa.ac.cn (R.L.); toptmz@163.com (M.T.); 18273175334@163.com (J.D.); 2College of Computer Science and Technology, Heilongjiang Institute of Technology, Harbin 150050, China; portzhang@foxmail.com; 3Beijing Focused Loong Technology Co., Ltd., Beijing 100086, China; 4Key Laboratory of Visual Perception and Artificial Intelligence of Hunan Province, College of Electrical and Information Engineering, Hunan University, Changsha 410082, China; binyang@hnu.edu.cn; 5College of Information and Communication, Harbin Engineering University, Harbin 150001, China; springrain@hrbeu.edu.cn

**Keywords:** live pigs, 3D camera, automatic weight measurement, growth curve, large-scale farming

## Abstract

**Simple Summary:**

The manual pig weighing method is time-consuming, labor-intensive, and stressful for pigs, and can even cause pig death. Efficiently and accurately obtaining live pig weights has always been the goal of pig producers. With the development of science and technology, using 3D depth images and artificial intelligence technology to estimate pig weight has gradually become a research hotspot. However, this technology is mostly in the research and development stage, with few practical applications. Therefore, our study verified the accuracy of a patrol automatic intelligent weighing system in practical production and explored its application effect in determining pig growth curves. The results indicated that the intelligent weighing system still has high accuracy in large-scale farming, and compared with the manual weighing method, the intelligent weighing method has a better goodness of fit. Automatic intelligent weighing systems (AIWS) can save manpower, improve production efficiency, and improve pig welfare. It is expected to replace manual weighing as the main weighing method in the future.

**Abstract:**

To verify the accuracy of AIWS, we weighed 106 pen growing-finishing pigs’ weights using both the manual and AIWS methods, respectively. Accuracy was evaluated based on the values of MAE, MAPE, and RMSE. In the growth experiment, manual weighing was conducted every two weeks and AIWS predicted weight data was recorded daily, followed by fitting the growth curves. The results showed that MAE, MAPE, and RMSE values for 60 to 120 kg pigs were 3.48 kg, 3.71%, and 4.43 kg, respectively. The correlation coefficient *r* between the AIWS and manual method was 0.9410, and R^2^ was 0.8854. The two were extremely significant correlations (*p* < 0.001). In growth curve fitting, the AIWS method has lower AIC and BIC values than the manual method. The Logistic model by AIWS was the best-fit model. The age and body weight at the inflection point of the best-fit model were 164.46 d and 93.45 kg, respectively. The maximum growth rate was 831.66 g/d. In summary, AIWS can accurately predict pigs’ body weights in actual production and has a better fitting effect on the growth curves of growing-finishing pigs. This study suggested that it was feasible for AIWS to replace manual weighing to measure the weight of 50 to 120 kg live pigs in large-scale farming.

## 1. Introduction

Body weight is an extremely important indicator in pig production and scientific research. Accurately obtaining the weight of pigs guides scientific breeding and improves economic benefits [1]. Direct weight measurement using a weighbridge is the most common manual method. This method can provide the most accurate weight, but it is time-consuming and labor-intensive, especially when the pig population is large. Besides, this method is difficult to achieve and is very stressful for pigs, affecting their growth and even causing pig death [2,3,4,5]. With the development of technologies such as image processing, image analysis, machine learning, deep learning, and computer vision, weighing methods have transformed from contact to noncontact [5]. Noncontact measurement is based on computer vision technology and collects animal image data by 2D and 3D cameras for weighing [6]. Combined with convolutional neural networks (CNN) [7] or recurrent convolutional neural networks (RNN) [8], it can extract pig body shape features quickly and estimate pig weight accurately [9]. For example, Jun et al. [10] developed an estimation model based on 2D images and accurately measured the weights of 70 to 120 kg pigs. Fernandes et al. [11] successfully predicted the body weight of nursery and finishing pigs via 3D computer vision. In addition, noncontact measurement can also achieve automatic, continuous, and real-time monitoring of live pigs through smart equipment, promote the development of precision livestock farming, and improve pig production efficiency [1,12].

In recent years, noncontact measurement has been a rising weight-measuring tool in precision livestock farming. To achieve efficient live pig weight measurement in large-scale farming, we introduce an automatic intelligent weighing system (AIWS). This system can measure the weight of large groups of pigs and may become one of the weight measurement tools in precision livestock farming. However, there is no report on the application of this AIWS in practical production. Therefore, to examine the accuracy of AIWS and its application in the growth curve fitting of pigs, two experiments were conducted on large-scale pig farming. We hope to promote AIWS application in large-scale farming and provide a reference method for weight measurement in precision livestock farming.

## 2. Materials and Methods

### 2.1. Animal Ethics

All animal care and handling followed protocols approved by the Animal Care and Use Committee of the Institute of Subtropical Agriculture, Chinese Academy of Sciences, under permit number IACUC # 201302 (Changsha, China).

### 2.2. Automatic Intelligent Weighing System (AIWS)

In this study, the AIWS uses an Intel RealSense D430 deep camera to collect 3D pig image data based on machine deep learning and the BotNet regression network to predict pig weight [13]. The input image resolution for the algorithm is 640 × 480, and the format is a 16-bit single channel. In the image, the value of each pixel represents the distance from that point to the camera. The image is captured using a depth camera and is processed through methods such as spatial three-dimensional point cloud denoising and filling before being converted into a two-dimensional format. This equipment consists of four main parts: a cloud service system, an artificial intelligence computing system, a data collection terminal, and an application side. The cloud service system is an essential part of the AIWS. Its main functions include setting up patrol robot running programs, monitoring patrol robot operation status, storing image and video data, etc. The artificial intelligence computing system is the core part of the AIWS and consists of pre-procession of the front and cloud platforms. The raw depth images are preprocessed by a series of preprocessing algorithms which include instance segmentation, distance independence, noise reduction, and rotation correction. Then, processed images are input into the algorithm model to predict the weight of the pig. A data collection terminal (depth camera), also called the patrol robot, is installed on an “I”-shaped track above the pig herd and 2010 ± 20 mm above the ground. It collects the back-depth images of free-moving pigs in the top-view automatically and moves them back and forth on the track (Figure 1). The system uploads the depth image data to the artificial intelligence computing system through the wireless network and estimates the weight of every pig. Then, the average weight per pen is calculated. Finally, the weight data are output to the mobile phone or computer application through the Internet for querying.

### 2.3. Animal and Experimental Design

#### 2.3.1. Exp.1. Accuracy Validation of AIWS

One hundred and six pen growing-finishing pigs (Duroc × Landrace × Large White) with body weights between 60 to 120 kg were selected, with 18 to 30 pigs per pen and 2794 pigs in total. During the experiment, we used a weighbridge to measure the weights of the pigs. The total body weight and total number of pigs per pen were recorded and then the average body weight per pen was calculated. Simultaneously, we recorded the corresponding predicted per-pen average weight of AIWS. Before the experiment began, the equipment was debugged, and the weight data recording started after the equipment ran stably. The patrols of the robot can be adjusted according to actual requirements. In this study, the robot patrolled every two hours and stopped to collect the back-depth image data of pigs in each pen for 2 min.

#### 2.3.2. Exp.2. Determination of Pig Growth Curve

A total of 360 growing-finishing pigs (Duroc × Landrace × Large White) with an average initial body weight of 49.72 ± 3.00 kg were randomly divided into six replicates with 60 pigs per replicate. All the trial pigs were manually weighed using a traditional weighbridge at the beginning of the experiment; subsequently, all pigs were weighed every two weeks. Body weights were recorded after every measurement. At the same time, the AIWS predicted weight of each replicate was recorded daily until the end of the experiment.

All the experimental pigs were uniformly fed and managed according to the large-scale farming farms.

### 2.4. Accuracy Evaluation Indicators

During the test, each pen pig was weighed individually in batches using a weighbridge, and the actual average weight of pigs was calculated. Additionally, we recorded the corresponding predicted average weight for each pen of the AIWS. The mean absolute error (MAE), mean absolute percentage error (MAPE), and root mean squared error (RMSE) were used to compare the actual average weight and predict average weight with reference to Ositanwosu et al. [14]. The calculation formula is as follows:(1)$$MAE=\frac{1}{N}\sum_{i=1}^{N}\left|Actuali-Predictedi\right|$$
(2)$$MAPE=\frac{1}{N}\sum_{i=1}^{N}\left|\frac{Actuali-Predictedi}{Actuali}\right|$$
(3)$$RMSE=\sqrt{\frac{1}{N}\sum_{i=1}^{N}{\left(Actuali-Predictedi\right)}^{2}}$$

### 2.5. Growth Models and Goodness of Fit

Logistic and Gompertz models were used to fit the growth curve of growing-finishing pigs. The parameters of the two models are shown in Table 1. The absolute growth rate was obtained by the first derivative of the model [15]. Two models were compared using the Akaike information criterion (AIC), Bayesian information criterion (BIC), coefficient of determination R^2^, and adjusted coefficient of determination R^2^*_aj_* [15,16], and the model with low AIC and BIC values was considered the best-fit model [17]. The equations of AIC and BIC were as follows:(4)$$AIC=n\times \ln(\frac{RSS}{n})+2k$$
(5)$$BIC=n\times \ln(\frac{RSS}{n})+k\ln\left(n\right)$$

In Equations (4) and (5), *n* denotes the number of samples, *k* is the number of parameters, and *RSS* is the residual sum of squares of the model [18,19].

### 2.6. Statistical Analysis

Excel 2016 software was used to calculate the MAE, MAPE, and RMSE. GraphPad Prism 8.2 software was used to analyze the correlation of two weight measurement methods. Origin 2022 was used for descriptive statistics first, and then Logistic and Gompertz models were used to fit the growth curve. According to the fitting results, the AIC and BIC values were calculated using Excel 2016. *p* < 0.05 was regarded as statistically significant, and *p* < 0.01 was regarded as extremely significant.

## 3. Results

### 3.1. Assessment of Weight Measurement Accuracy

The detailed data of the two weight measurement methods for 60 to 120 kg growing-finishing pigs are available in Table A1. As shown in Table 2, the MAE value was 3.48 kg, the MAPE value was 3.71%, and the RMSE value was 4.43 kg.

### 3.2. Correlation Analysis

The results of the correlation analysis of the two weight measurement methods are provided in Figure 2. The correlation coefficient of the two weight measurement methods for 60 to 120 kg growing-finishing pigs was *r* = 0.9410, and the determination coefficient was R^2^ = 0.8854. An extremely significant correlation (*p* < 0.001) was observed between the two weighing methods.

### 3.3. Growth Curve Fitting

The detailed data of days and weight of pigs by different measurement methods are shown in Table A2. The fitting results of the two models are listed in Table 3. In both weighing methods, the *A* values were 153.23 kg and 186.89 kg in the Logistic model, the *B* values were 21.28 and 18.68, and the *k* values were 0.0218 and 0.0178, respectively. The *A* values were 200.90 kg and 345.33 kg for the Gompertz model, the *t_c_* values were 137.08 and 203.63, and the *k* values were 0.0111 and 0.0069, respectively. The growth curves of 50 to 110 kg growing-finishing pigs are presented in Figure 3. At this stage, the growth curves of pigs showed tilted right J-shaped increases. All curves showed the same increasing trend. The curves of the absolute growth rate are shown in Figure 4. According to Figure 4, the growth rate of growing-finishing pigs first increased and then decreased.

### 3.4. Goodness of Fit

From Table 4, in the Logistic model, the R^2^ values were 0.9989 and 0.9970 for the manual and AIWS weighing methods, and the R^2^_aj_ values were 0.9981 and 0.9969, respectively. The AIC and BIC values of the AIWS were much lower than the manual method. In the Gompertz model, R^2^ = 0.9990, R^2^ = 0.9967, R^2^_aj_ = 0.9984, and R^2^_aj_ = 0.9966 for the manual and AIWS weighing methods, respectively. Similarly, the AIC and BIC values of the AIWS were much lower than those of the manual method.

Comparing the four fitting results, the best-fit model was the Logistic model by AIWS. The expression of the best-fit model is shown in Table 5, achieving a maximum growth rate at an inflection point of 164.46 d, and the maximum growth rate was 831.66 g/d.

## 4. Discussion

The camera is a direct factor affecting the accuracy of noncontact weight measurement. Noncontact weight measurement usually uses pig images collected by 2D and 3D cameras for weighing [4,19]. Although the accuracy was high, there were some problems with the use of 2D image data and camera systems in large-scale farming. For example, the data quality of a 2D camera system depends largely on the illumination conditions [20]. Suboptimal lighting conditions can significantly reduce estimation accuracy [21]. The data extracted from images captured in different environments may interfere with image processing and analysis, leading to incorrect results [22]. Compared to 2D camera systems, the 3D camera has more advantages. Current 3D camera systems have their own light sources or use infrared, making them robust in different lighting environments [20]. Three-dimensional images (depth images) are not affected by light [19], and 3D imaging also reduces the influence of light on the weight estimation accuracy [23]. One of the greatest factors affecting the estimation model accuracy for pig body weight is the depth camera accuracy [24]. The depth camera accuracy is influenced by three factors: distance from the camera to the target, temperature, and target color [25,26]. Therefore, understanding the depth camera accuracy used and the effects of different environmental conditions on its accuracy is crucial for accurate body weight estimation. In this experiment, the distance from the camera to the target was the optimal distance obtained through extensive experiments which was fixed and unchanged, so it had little impact on weight measurement. However, a few larger absolute percentage errors may be related to the color of the pigs’ bodies. If a pig’s body is dark-colored or the surface is dirty, the measurement values will deviate [25], which in turn affects the weight estimation accuracy. In addition, the posture of pigs may also affect the weight estimation accuracy. The weight estimation model in our study was established based on the back image of standing pigs. The back posture changed when the pig lay on its stomach or side, resulting in differences between the collected depth image data and the model dataset, ultimately leading to larger absolute percentage errors.

In addition to depth cameras, predictive networks are also crucial for the accuracy of estimation results. ANNs and convolutional neural networks (CNNs) in conjunction with image processing improved prediction accuracy [12]. As machine-learning algorithms, ANNs have achieved enormous success in a wide variety of fields [27]. CNN is a special type of ANN that has been optimized for input data with grid patterns such as images or videos and has excellent performance in image analysis data [12,28]. It can extract body shape features and estimate pig weight and body size quickly and accurately with simple automated preprocessing of 3D images [12]. Previous studies reported that CNNs trained once can function in real time and accurately identify pigs with an accuracy of approximately 96.7% [29]. In addition, Meckbach et al. [30] reported that providing only depth images and the related weight to the CNN was sufficient to accurately predict the body weight of 20 to 133 kg pigs (R^2^ > 0.97). In this weight measurement system, a regression network was built based on BotNet to predict the weight precisely. It was designed so that a dual branch of 3 × 3 convolution and MHSA replaces a single 3 × 3 convolution of the fourth block in ResNet, using 3D images as input. A high-accuracy and strong universality weight measurement model was obtained based on deep learning and considerable dataset training [13]. Then, cascading deconvolution layers and atrous convolution layers were used to improve the mask generation branch and solve the problem of low-resolution feature maps in the mask branch [31]. In the weight measurement of live pigs, this method optimized the Track R-CNN to output more accurate masks than the original network. In this experiment, the MAE, MAPE, and RMSE were 3.48 kg, 3.71%, and 4.43 kg, respectively. This result was close to the research (R^2^ = 0.65, MAE = 1.85 kg, MAPE = 1.68%, RMSE = 5.74 kg) of Chen et al. [1] who constructed a multilayer RBF neural network (deep neural network) model that automatically predicts the weight of live pigs. The RMSE was lower than the optimal prediction model of the multilayer perceptron neural network (MLP-NN) that was developed by Ositanwosu et al. [14]. This indicated that our prediction model is more precise and that the prediction results are more accurate. Furthermore, the correlation coefficient *r* of the two weight measurement methods was 0.9410, R^2^ was 0.8854, and there was a highly significant correlation between the two methods (*p* < 0.001). These results suggested that this AIWS could replace manual measurement to measure the weight of 60 to 120 kg pigs in large-scale farming.

A pig’s weight and growth rate are important factors in its production [32]. Fitting growth curves of pigs can predict pig growth, frame proper feeding plans, maximize utilization of the meat growth potential, and estimate the optimal slaughter weight for higher production and economic benefits [33,34]. Logistic and Gompertz are two commonly used and classical models in animal growth curve fitting, and they have a better fitting effect. Shen et al. [35] used three nonlinear growth models (Logistic, Gompertz, and Von Bertalanffy) to describe the growth characteristics of Liangshan pigs, established a growth curve model for Liangshan pigs, and estimated the maximum growth rate of Liangshan pigs. Studies have reported that the logistic model was the best-fitting model for longitudinal testicular volume in Nellore bulls [15]. The Gompertz model has a better-fit effect on the growth curve of partridges and has a higher R^2^ and R^2^_aj_ and a lower AIC and BIC [33]. Hoang et al. [36] showed that the Gompertz model was the most suitable model for describing the growth curve of Mia chickens. However, unlike these reports, Mata Estrada et al. [37] compared four nonlinear growth models and found that the Von Bertalanffy growth model had the best-fitting effect on Creole chickens in Mexico. It can be concluded that different varieties and different animals are suitable for different growth models. Finally, with the previous studies, we chose the Logistic and Gompertz models to fit the growth curve of 50 to 110 kg three-crossbred pigs (Duroc × Landrace × Large White). The results of our study revealed that the AIC and BIC of the AIWS weighting method were much lower than the manual method, and a better fit was indicated by a lower AIC and BIC [38,39], suggesting that the AIWS weight estimation method is superior to manual weight measurement in growth curve fitting. In AIWS weight estimation, both R^2^ and R^2^_aj_ of the Logistic and Gompertz models were 0.997, and the AIC and BIC of the Logistic model were lower than those of the Gompertz model. Therefore, we considered that the logistic model obtained by the machine weighing method was the best-fitting model. In this experiment, the age at the inflection point of 50 to 110 kg three-crossbred pigs was 164.46 d and the body weight at the inflection point was 93.45 kg.

In this study, the growth curve for 50 to 110 kg pigs showed a tilted right J-shaped increase because the growth rate of pigs gradually decreased after reaching the inflection point, and the curve tended to flatten. In addition, the farms began to sell and slaughter pigs after 180 days when their body weight reached approximately 110 kg to achieve maximum breeding benefits. Our experiment also ended with the sale and slaughter of pigs. Consequently, the growth curve of pigs did not reach a stable period and was not S-shaped. The maximum growth rate was estimated using growth curves [40] and calculated using the first derivative of the growth curve absolute growth rate (AGR) [41]. We took the first-order derivative of two models using two weighing methods to obtain the absolute growth rate, as shown in Figure 4. With aging, the AGR first increased and then decreased, reaching its maximum growth rate at the inflection point. The maximum growth rate of the best-fit model was close to the method of manual weighing. It was 831.66 g/d. This result is higher than that of Liangshan pigs (455.43 g/d) [42], and it also confirmed that three-crossbred pigs grew faster than local pigs. The growth rate and the daily gain decreased after the inflection point, and the feed conversion ratio also decreased accordingly. It is also the main reason that the farms sell and slaughter pigs after 180 days of age when their body weight reaches around 110 kg. Therefore, the logistic model built by the AIWS weighing method not only conformed to the growth and development rules of pigs but also conformed to practical production. For these reasons, it is feasible for AIWS to replace manual weighing in fitting pig growth curves, and continuous weight data on pigs had a better fitting effect.

## 5. Conclusions

In conclusion, our results suggest that the automatic intelligent weighing system (AIWS) has high weight measurement accuracy and good stability. AIWS can replace manual weighing to accurately measure the weight of 50 to 120 kg live pigs in large-scale farming. In addition, it also had a good application effect in fitting the growth curve of live pigs. The AIC and BIC values were both the lowest at −174.1373 and −167.2659, respectively. Therefore, against the background of big data, the real-time, continuous, and automatic intelligent weight estimation system has broad development prospects, and it is a very promising weight measurement tool for precision farming in the future.

## Figures and Tables

**Figure 1 animals-14-01614-f001:**
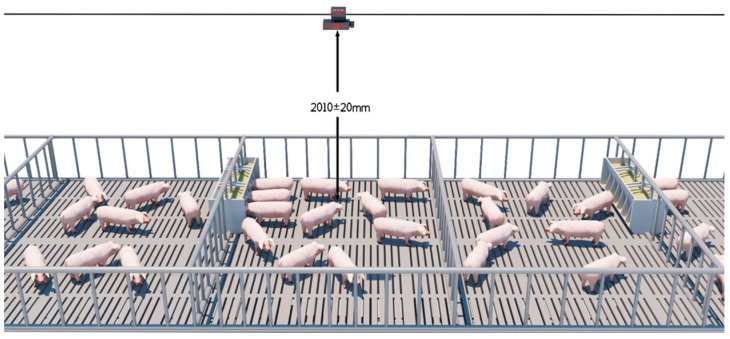
Schematic diagram of data collection. The distance between the depth camera and the ground is 2010 ± 20 mm.

**Figure 2 animals-14-01614-f002:**
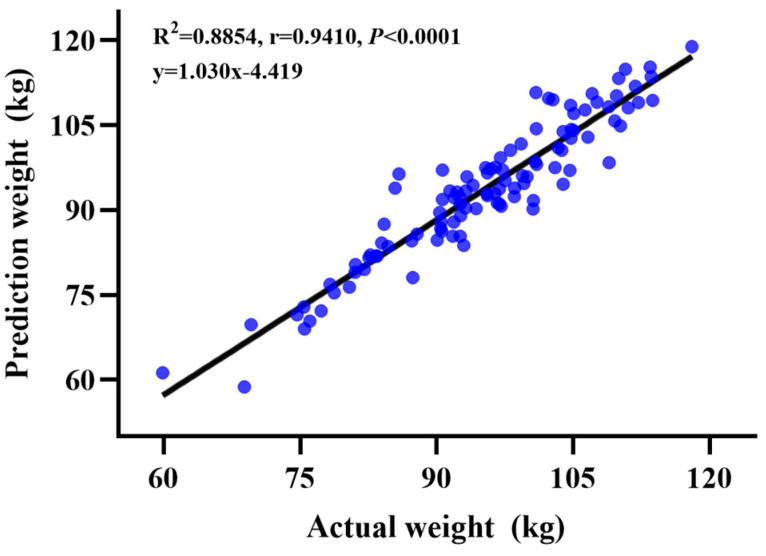
Correlation analysis of two weight measurement methods.

**Figure 3 animals-14-01614-f003:**
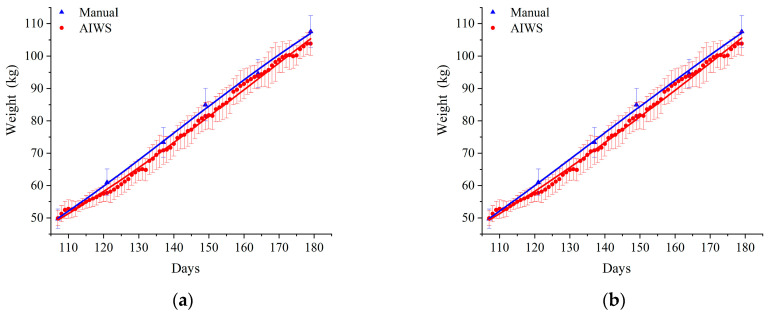
The growth curves of two models for 50–110 kg pigs in different weighing methods: (**a**) Logistic model, (**b**) Gompertz model. AIWS = automatic intelligent weighing system.

**Figure 4 animals-14-01614-f004:**
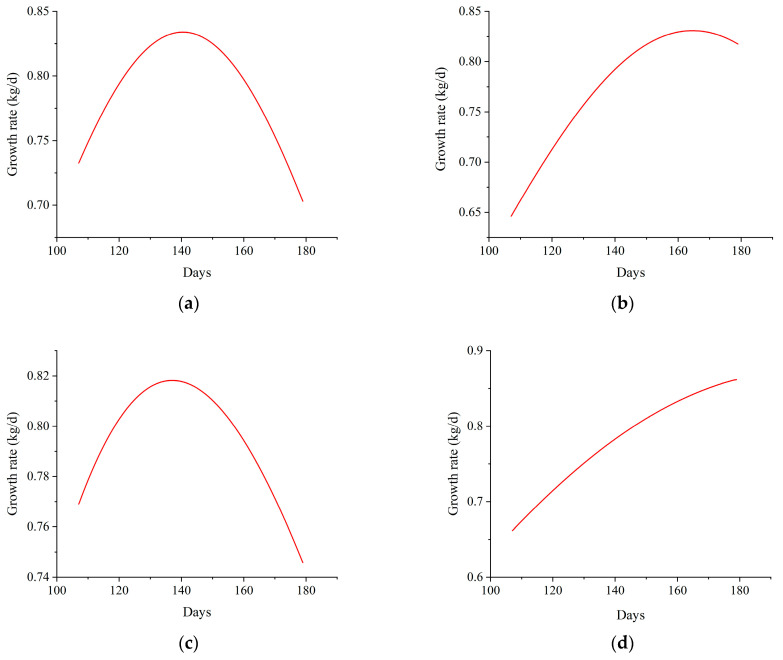
The absolute growth rate curve of the first derivative. (**a**) Growth rate of Logistic model in manual weighing, (**b**) Growth rate of Logistic model in AIWS weighing, (**c**) Growth rate of Gompertz model in manual weighing, (**d**) Growth rate of Gompertz model in AIWS weighing.

**Table 1 animals-14-01614-t001:** Growth curve models and characteristic parameters.

Model	Equation	Parameters	Age at IP (d)	Weight of IP (kg)	MGR (g/d)
Logistic	W*_t_* = *A/*(1 *+ Be^−kt^*)	*A*, *B*, *k*	*(lnB)/k*	*A*/2	*Ak*/4
Gompertz	W*_t_* = *Aexp*(*−exp*(*−k*(*t − t_c_*)))	*A*, *k*, *t_c_*	*t_c_*	*A/e*	*Ak/e*

Note: IP = inflection point; MGR = maximum growth rate; *W_t_* = body weight (kg) at day *t*; *A* = asymptotic weight (in Logistic) or average weight at maturity (in Gompertz); *B* = integration constant; *k* = maturity rate; *t_c_* = inflection point age.

**Table 2 animals-14-01614-t002:** The MAE, MAPE, and RMSE values for 60 to 120 kg growing-finishing pigs.

Weight Interval (kg)	MAE (kg)	MAPE (%)	RMSE (kg)
60 to 120	3.48	3.71	4.43

Note: MAE = mean absolute error; MAPE = absolute percentage error; RMSE = root mean squared error.

**Table 3 animals-14-01614-t003:** The fitting parameter values of two models for 50 to 110 kg growing-finishing pigs.

Model	Logistic		Gompertz	
Weighing Method	Manual	AIWS	Manual	AIWS
*A*	153.23 ± 13.40	186.89 ± 11.32	200.90 ± 31.92	345.33 ± 59.94
*B*	21.28 ± 2.45	18.68 ± 0.31	-	-
*t_c_*	-	-	137.08 ± 14.94	203.63 ± 23.75
*k*	0.0218 ± 0.0021	0.0178 ± 0.0008	0.0111 ± 0.0019	0.0069 ± 0.0008
Age at IP (d)	140.26	164.46	137.08	203.32
*W* of IP (kg)	76.62	93.45	73.91	127.04
MGR (g/d)	835.1	831.66	820.1	876.57

Note: *A* = asymptotic weight (in Logistic) or average weight at maturity (in Gompertz); *B* = integration constant; *t_c_* = inflection point age; *k* = maturity rate; IP = inflection point; *W* = weight; MGR = maximum growth rate; AIWS = automatic intelligent weighing system.

**Table 4 animals-14-01614-t004:** The goodness of fit criteria values of Logistic and Gompertz models in different weighing methods.

Model	Weighing Method	AIC	BIC	R^2^	R^2^*aj*
Logistic	Manual	−15.3798	−16.0045	0.9989	0.9981
	AIWS	−174.1373	−167.2659	0.9970	0.9969
Gompertz	Manual	−16.2913	−24.4122	0.9990	0.9984
	AIWS	−165.6111	−158.7398	0.9967	0.9966

Abbreviations: AIC = Akaike information criterion; BIC = Bayesian information criterion; AIWS = automatic intelligent weighing system; R^2^ = coefficient of determination; R^2^*_aj_* = adjusted coefficient of determination.

**Table 5 animals-14-01614-t005:** The expression of the best-fit model and growth inflection point.

Weighing Method	Model	Expression	Age at IP (d)	Weight of IP (kg)	MGR4 (g/d)
AIWS	Logistic	*W_t_* = 186.89/(1 + 18.68*e*^−0.0178*t*^)	164.46	93.45	831.66

Abbreviations: AIWS = automatic intelligent weighing system; *W_t_* = body weight (kg) at day *t*; IP = inflection point; MGR = maximum growth rate.

## Data Availability

All data are included in the article.

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
