# Peer review of "Improving Efficiency: Automatic Intelligent Weighing System as a Replacement for Manual Pig Weighing"

_animals, 2024, doi:10.3390/ani14111614_

Round 1

Reviewer 1 Report

Comments and Suggestions for Authors The main question raised in this article is: Does the development of an automatic intelligent weighing sys- 65 tem (AIWS) can measure the weight of large groups of pigs? The relevant part is the development of the automatic intelligent weighing system but they need to better describe this system.   There are some automatic intelligent weighing systems developed but the novelty of this one is for a large group of pigs.   Pg 79 -89. - Please describe the system with more details, for example: authors say about an artificial intelligence computing system, how is this system composed?  Authors say "The system uploads the depth image data to the predic- 86 tion model through the wireless network and estimates the weight of every pig, then the 87 average weight of per pen was calculated."How does the system estimate the weights?  

Abstract: some methods are missing

Methods: The Automatic Intelligent Weighing System (AIWS) is not well described.

Discussion: You must discuss the results not do a literature review of death cameras

  The conclusions are ok (all main questions were addressed). The references are appropriate and cover a large part of the publications within the studied area   I don't think "Table A1. Detailed data of two weight measurement methods for 60-120kg growing-finishing pigs."is needed

Author Response

Thank you very much for taking the time to review this manuscript. The comments and suggestions definitely helped us to improve the quality of the manuscript. We have revised the manuscript in which major changes are highlighted in red/bule, the detailed corrections are listed below. 

Comments 1: Does the development of an automatic intelligent weighing sys- 65 tem (AIWS) can measure the weight of large groups of pigs?

Response 1: Yes, it does. AIWS can measure the weight of large groups of pigs in large-scale farming.  

Comments 2: The relevant part is the development of the automatic intelligent weighing system but they need to better describe this system.

Response 2: Thanks for your advice. We agree with this comment. We have added a description of this system. The change can be found in the manuscript lines 93-100.

Comments 3: Please describe the system with more details, for example: authors say about an artificial intelligence computing system, how is this system composed?

Response 3: Thanks. We have added a description of cloud service and artificial intelligence computing system. It can be found in the manuscript lines 93-100.

Comments 4: How does the system estimate the weights?

Response 4: First, the patrol robot collects the depth images of pigs and uploads the depth image data to the artificial intelligence computing system through the wireless network. Second, the artificial intelligence computing system processed the raw depth images through preprocessing algorithms. Then, processed images are input into the algorithm model. After model training, the weight of pigs can be predicted.

Comments 5: Abstract: some methods are missing.

Response 5: Agree. We have made this correction in the abstract as your suggestion. The change can be found in the manuscript line 36-38.

Comments 6: Methods: The Automatic Intelligent Weighing System (AIWS) is not well described.

Response 6: We have revised it as your suggestion. It can be found in the manuscript lines 93-100.

Comments 7: Discussion: You must discuss the results not do a literature review of depth cameras.

Response 7: Thanks for your suggestion. We have revised it and deleted partial content. The change can be found in the manuscript lines 232-244.

Comments 8: I don't think "Table A1. Detailed data of two weight measurement methods for 60-120kg growing-finishing pigs. " is needed.

Response 8: Thanks for your suggestion. However, the Table A1 is an additional material and not part of the main text content. In order to meet the needs of more readers, we would like to keep it.

Reviewer 2 Report

Comments and Suggestions for Authors

A manuscript submitted for review on Improving efficiency: automatic intelligent weighing system as a replacement for manual pig weighing examines a very important issue for farmers, namely determining the weight of pigs. This question is very important, because the health status of the pigs is determined based on this indicator, as well as the conditions of their breeding and feeding. With the alternative method of weighing the live weight of the pigs, it is possible to experience stress that affects the whole process, and it is also labor intensive. All this allows me to assess the topicality of the topic as extremely high and important for practice.

I have the following notes for the authors:

Line 39 and line 106 have spelling mistakes.

In conclusion, it is claimed that the method is without any stress for the pigs. This cannot be claimed because the authors have not presented similar results (of behavior or stress hormones) this statement of theirs is only based on assumptions and cannot be included in the conclusion.

Based on everything described, I can claim that the manuscript has a very high scientific and practical value and can be accepted after removing these notes.

Author Response

Thank you very much for taking the time to review this manuscript. The comments and suggestions definitely helped us to improve the quality of the manuscript. We have revised the manuscript in which major changes are highlighted in red/bule, the detailed corrections are listed below. 

Comments 1: Line 39 and line 106 have spelling mistakes.

Response 1: Thank you for pointing out the mistakes. We have corrected the mistakes.

Comments 2: In conclusion, it is claimed that the method is without any stress for the pigs. This cannot be claimed because the authors have not presented similar results (of behavior or stress hormones) this statement of theirs is only based on assumptions and cannot be included in the conclusion.

Response 2: Thank you very much for your careful comments and valuable suggestion. We have removed this note. The change can be found in the manuscript line 342.

Reviewer 3 Report

Comments and Suggestions for Authors

The research is relevant and can find application in practical use. A sufficient number of scientists around the world are engaged in this work, which confirms the relevance of the research. The scientific novelty of the research is the development of a method for non-contact assessment of the weight of pigs.

In general, I would like to note the high quality of the work done, but also pay attention to some comments:

Point 1: It is necessary to expand the literature review for this study as 6 sources is not a sufficiently representative number. The experience of other scientists in this topic should be considered, not only in creating similar systems, but also in the use of neural network algorithms and technical measurement tools.

Point 2: The chapter on materials and methods did not indicate the parameters of the sample used to train the neural networks.

Point 3: Describe in more detail the theoretical and practical significance of your research, and it is also necessary to clarify specific examples of applying research results in practice.

Point 4: A block algorithm for the operation of the developed system should be presented.

Point 5: The abstract should avoid vague terms such as “high precision”, “can improve production efficiency”, “improve pig welfare”. Specific results on the accuracy of the developed system should be given. How specifically will your research improve pig welfare?

Point 6: Provide calculations of the economic efficiency of the device.

Point 7: In output 1, provide a specific result about the accuracy of your system. In line 325 replace “...good application effect...” with specific numerical values.

The work can be published after revision in accordance with the recommendations.

Author Response

Thank you very much for taking the time to review this manuscript. The comments and suggestions definitely helped us to improve the quality of the manuscript. We have revised the manuscript in which major changes are highlighted in red/bule, the detailed corrections are listed below.

Point 1: It is necessary to expand the literature review for this study as 6 sources is not a sufficiently representative number. The experience of other scientists in this topic should be considered, not only in creating similar systems, but also in the use of neural network algorithms and technical measurement tools.

Response 1: Thank you very much for your careful comments and valuable suggestion.We have revised them in the manuscript. The changes can be found in the manuscript lines 60-66.

Point 2: The chapter on materials and methods did not indicate the parameters of the sample used to train the neural networks.

Response 2: Thanks for your suggestion. We have revised them in the manuscript. The changes can be found in the manuscript lines 87-91.

Point 3: Describe in more detail the theoretical and practical significance of your research, and it is also necessary to clarify specific examples of applying research results in practice.

Response 3: Thanks for your suggestion. We have revised them in the introduction part.

Point 4: A block algorithm for the operation of the developed system should be presented.

Response 4: Thanks for your suggestion. We have described it in the discussion part. It can be found in lines 272-281.

Point 5: The abstract should avoid vague terms such as “high precision”, “can improve production efficiency”, “improve pig welfare”. Specific results on the accuracy of the developed system should be given. How specifically will your research improve pig welfare?

Response 5: Thanks for your suggestion. This cannot be claimed because we have not presented similar results. We have removed this note. The change can be found in the manuscript line 344.

Point 6: Provide calculations of the economic efficiency of the device.

Response 6: Thanks for your suggestion. We can’t provide it because we have not done similar test. We have removed this note. The change can be found in the manuscript line 344.

Point 7: In output 1, provide a specific result about the accuracy of your system. In line 325 replace “...good application effect...” with specific numerical values.

Response 7: Thanks for your suggestion. We have revised them in the manuscript. The change can be found in the manuscript line 345-346.